# A Novel Evaluation Framework for Image Inpainting via Multi-Pass Self-Consistency

## Abstract

Image inpainting aims to restore missing regions of corrupted images by utilizing the available unmasked content while ensuring consistency and fidelity. In scenarios where limited information is available, determining a unique optimal solution for a given inpainting case becomes challenging. However, existing assessment approaches predominantly rely on the availability of corresponding unmasked images, which introduces potential biases toward specific inpainting solutions. To address this disparity, we propose a novel evaluation framework that leverages the power of aggregated multi-pass image inpainting. Our self-supervised metric offers exceptional performance in scenarios with or without unmasked images. Rather than solely relying on similarity to the original images in terms of pixel space or feature space, our method prioritizes intrinsic self-consistency. This allows us to explore diverse and viable inpainting solutions while mitigating biases. Through extensive experimentation on multiple baselines, we demonstrate the strong alignment of our method with human perception, which is further supported by a comprehensive user study.

## 1 Introduction

Image inpainting (Bertalmio et al., 2000) is a long-standing topic in computer vision, aiming to fill in missing regions of corrupted images with semantically consistent and visually convincing content. Recent advancements in image inpainting have brought benefits to various applications, including image editing (Jo & Park, 2019), photo restoration (Wan et al., 2020), and object removal (Yildirim et al., 2023). Despite the promising results achieved by state-of-the-art approaches, effectively inpainting complex image structures and large missing areas remains a challenging task.

Due to the inherently ill-posed nature of the image inpainting problem, reliable evaluation metrics are lacking. Evaluation metrics commonly used for assessing inpainting performance can be categorized into two groups. The first group involves direct comparisons of similarity between paired original and restored images, either in the pixel space or the embedded feature space. Examples of such metrics include Mean Squared Error, Peak Signal-to-Noise Ratio, Structural Similarity Index (Wang et al., 2004), and Learned Perceptual Image Patch Similarity (Zhang et al., 2018). The second group of metrics measures the distance between the distributions of inpainted images and the original images, such as the Frechet Inception Distance (Heusel et al., 2017). However, these metrics require comparison with unmasked images, which may not always be available in practical scenarios. Thus, there is a need for a metric that can be based solely on the inpainted images themselves.

Another concern relates to the potential bias introduced by the aforementioned metrics. Figure 1 serves as an illustrative example to highlight this issue. In practical scenarios, the mask representing the corrupted area within an image often covers a significant portion, posing a formidable challenge in accurately predicting the content hidden by the mask. Moreover, the content within the corrupted region may have multiple plausible solutions, which is a common occurrence in real-world images. As depicted in Figure 1, it is impossible to determine the exact height and pattern of the rock within the masked area, making all plausible outcomes acceptable. More detailed discussions are provided in Figure 3 and Section 3.3. Consequently, directly utilizing unmasked images as the basis for evaluating inpainting methods can result in biased assessments.

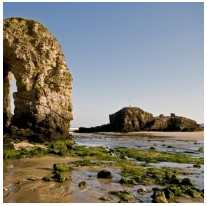 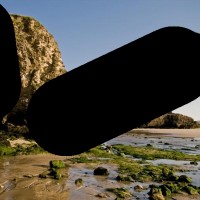 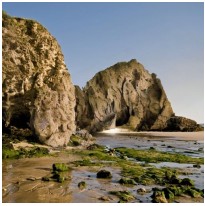 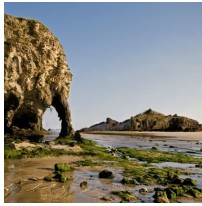 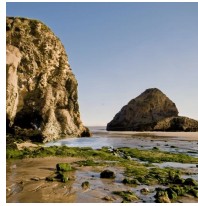

(a) Original Image    (b) Masked Image    (c) Inpainted 1    (d) Inpainted 2    (e) Inpainted 3

Figure 1: An example showcases the potential variations in inpainted results for a single image. The presence of a large masked area, which may encompass crucial content that cannot be accurately restored by inpainting methods, leads to inpainted images with multiple possible layouts. Comparing the inpainted images directly to the original images can introduce bias into the evaluation process.

One potential approach to evaluating inpainting methods is to assess their understanding of the content in both the damaged images and the content they generate themselves. This concept aligns with the words of the esteemed scientist Richard Feynman, who stated, "*What I cannot create, I do not understand*". An exemplary inpainting method should demonstrate *self-consistency in its inpainted images*. This implies that the inpainted content in the missing regions can generate content in the unmasked regions. If we re-inpaint the inpainted images, these re-inpainted images should be identical to the original inpainted images. By achieving such a high level of consistency, the inpainting method can demonstrate its profound understanding of the generated content.

Building upon this hypothesis, we present a novel framework for unbiased evaluation of image inpainting methods. Our proposed framework involves the selection of an inpainting method, followed by the application of a random inpainting method using multiple new masks to re-inpaint the inpainted images. To ensure context-level stability between the re-inpainted images and the original inpainted images, we employ a patch-wise mask, thereby enhancing the multi-pass stability of the evaluation process. This innovative benchmark enables the evaluation of inpainting methods without the need for uncorrupted images, offering valuable insights into the image inpainting task. Extensive experimentation validates that our proposed benchmark closely aligns with human evaluation, eliminating the reliance on unmasked image comparisons.

## 2 RELATED WORKS

In this section, we present an overview of the image inpainting task and highlight the state-of-the-art deep image inpainting methods. Additionally, we delve into the realm of perceptual metrics for image inpainting, which constitutes the focus of this paper.

**Image Inpainting** The field of image inpainting has been under development for several decades since the formal proposal of the task by Bertalmio *et al.* (Bertalmio et al., 2000). Traditional image inpainting approaches can be categorized into two main types: diffusion-based and exemplar-based methods. Diffusion-based methods (Richard & Chang, 2001; Tschumperlé, 2006; Li et al., 2017; Daribo & Pesquet-Popescu, 2010) fill the missing region by smoothly propagating image content from the boundary to the interior of the region. Exemplar-based approaches (Efros & Leung, 1999; Efros & Freeman, 2001; Le Meur & Guillemot, 2012; Criminisi et al., 2004; Barnes et al., 2009; Ružić & Pižurica, 2014) search for similar patches in undamaged regions and leverage this information to restore the missing part.

The emergence of deep learning has prompted researchers to propose numerous deep models to enhance inpainting performance. Nazeri *et al.* (Nazeri et al., 2019) introduced a two-stage adversarial model that first generates hallucinated edges and then completes the image. Yu *et al.* (Yu et al., 2019) devised gated convolution and a patch-based GAN loss for free-form mask settings. Zhao *et al.* proposed a co-modulated generative adversarial network architecture for image inpainting, embedding both conditional and stochastic style representations. Suvorov *et al.* (Suvorov et al., 2022) utilized fast Fourier convolutions (FFCs) and achieved remarkable performance in handling large missing areas and high-resolution images. Rombach *et al.* (Rombach et al., 2022) introduced latent diffusion models and applied them to image inpainting. Despite the promising results obtained

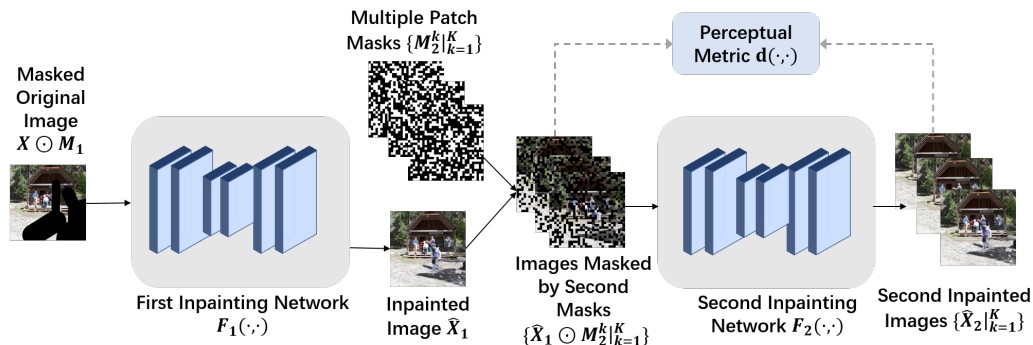

Figure 2: Overview of our proposed image inpainting metric. We incorporate a multi-pass approach to enhance evaluation stability by iteratively re-inpainting the inpainted images using multiple patch masks. This iterative process allows us to calculate the perceptual metric between the inpainted images and the corresponding re-inpainted images, thereby capturing the consistency and fidelity of the inpainting method.

by these works, achieving high-fidelity completed images with self-consistent context remains a challenge, especially when dealing with complex structures and large irregular missing areas.

**Perceptual Metrics**  Commonly used metrics for evaluating the performance of image inpainting can be classified into two categories. The first category involves direct comparisons of similarity between paired original and restored images in either the pixel space or the embedded feature space. Examples of such metrics include Mean Squared Error (MSE), Learned Perceptual Image Patch Similarity (LPIPS) (Zhang et al., 2018), Structural Similarity Index (SSIM) (Wang et al., 2004), and Peak Signal-to-Noise Ratio (PSNR). However, considering that the inpainting result is not uniquely determined by the known part of an image, the restored portion is not necessarily required to be identical to the original image. These metrics confine the solutions to a subset of all feasible options, potentially introducing biases and overfitting issues.

The second category of metrics measures the distance between the distributions of inpainted images and the original images. Metrics such as the Frechet Inception Distance (FID) (Heusel et al., 2017) and Paired/Unpaired Inception Discriminative Score (P/U-IDS) (Zhao et al., 2021) quantify the perceptual fidelity of inpainted images by assessing their linear separability in the deep feature space of Inception models (Szegedy et al., 2016). However, in certain scenarios, it may not be feasible to obtain a sufficiently large dataset for accurately computing the distribution distance. Thus, the applicability of these metrics can be limited.

Our approach distinguishes itself from these methods by achieving reliable image quality assessment using a single image without the need for an unmasked image reference. This allows for a self-consistency metric that ensures the context of the inpainted image remains consistent throughout the restoration process.

## 3  THE PROPOSED BENCHMARK

In this section, we first introduce the image inpainting task and then present our proposed evaluation framework. Subsequently, we discuss the bias introduced by previous evaluation framework and demonstrate how our proposed benchmark can alleviate this bias.

### 3.1  NOTATIONS

Image inpainting is a task that aims to restore missing regions in corrupted images, ensuring both visual coherence and semantic consistency. Let $\mathbf{X} \in \mathbb{R}^{w \times h \times 3}$ denote the *original image* with width $w$ and height $h$, and $\mathbf{M}_1 \in \{0, 1\}^{w \times h}$ represent the corresponding binary *mask*, where 1 (*resp.*,

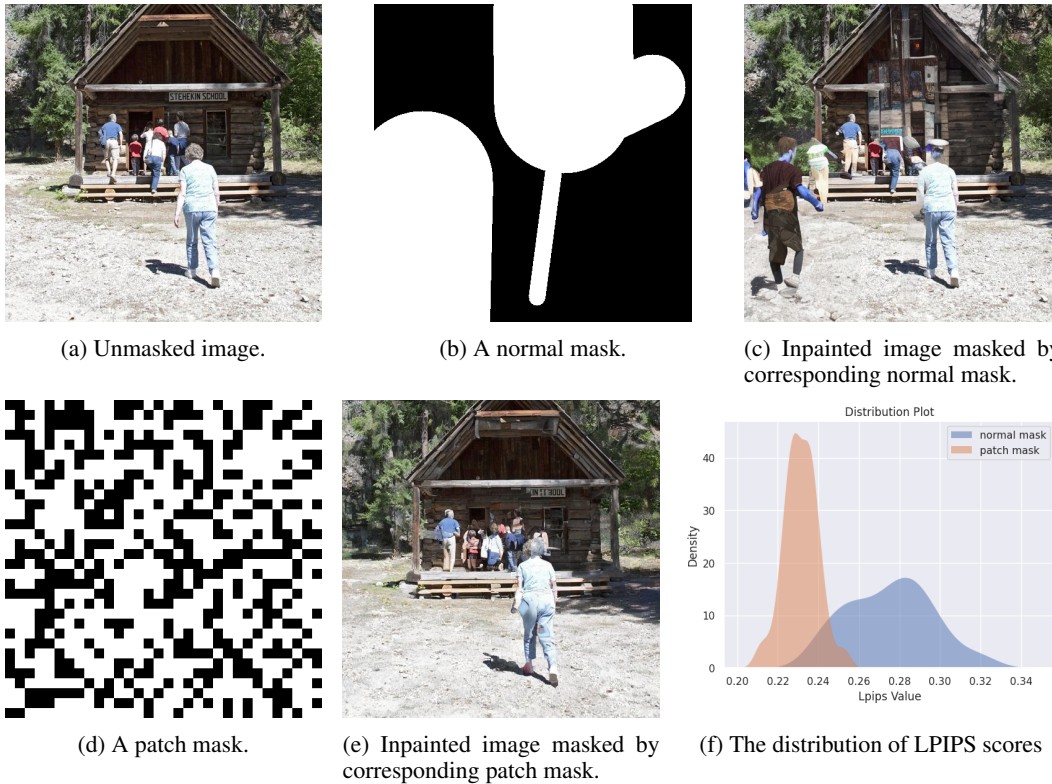

(a) Unmasked image.

(b) A normal mask.

(c) Inpainted image masked by corresponding normal mask.

(d) A patch mask.

(e) Inpainted image masked by corresponding patch mask.

(f) The distribution of LPIPS scores

Figure 3: Comparison of inpainted images masked by normal mask and patch mask. Figure 3a 3b 3c 3d 3e show image examples under different settings. Figure 3f shows the distribution of LPIPS scores with different types of masks (normal or patch masks) relative to the original image. For each type of mask, we use 100 different random seeds using StableDiffusion with the same mask and the same original image.

0) indicates masked (*resp.*, unmasked) pixels. We also call $\mathbf{M}_1$ as the *first mask*. The objective of the image inpainting task is to restore the damaged image $\mathbf{X} \odot \mathbf{M}_1$, where $\odot$ denotes element-wise product.

Our proposed evaluation framework aims to assign a score to an inpainting method $F_1(\cdot, \cdot)$ (*a.k.a.*, the *first inpainting network*), which takes $\mathbf{X} \odot \mathbf{M}_1$ and $\mathbf{M}_1$ as input and outputs an inpainted image $\hat{\mathbf{X}}_1 = F_1(\mathbf{X} \odot \mathbf{M}_1, \mathbf{M}_1)$. This inpainted image is referred to as *the first inpainted image*.

## 3.2 THE PROPOSED FRAMEWORK

The evaluation of image inpainting involves both visual quality of the generated images and appropriateness of the content. Similarly, inpainting networks rely on both visual appearance and global context to determine what to inpaint. If either the appropriateness or fidelity of one aspect is compromised, or if there's a lack of overall consistency, the model tends to produce less natural and more chaotic inpaintings. A natural image or an ideal inpainted image inherently possesses high intrinsic consistency, due to myriad interconnections present in the real world, such as physical laws or the joint probability distribution of various image elements. Such consistency provides clear guidance on the following inpainting. On the other side, unnatural images or poorly inpainted images are not seen in the training dataset of any inpainting networks and tend to get low performance as a consequence.

Motivated by the above perspective, we propose our evaluation framework for image inpainting that mitigates bias through multi-pass self-consistency. Within this framework, we introduce an additional binary mask $\mathbf{M}_2 \in \{0, 1\}^{w \times h}$ (*a.k.a.*, the *second mask*) and an inpainting method $F_2(\cdot, \cdot)$

---

**Algorithm 1** Random Mask Generator

---

**Require:** Image to be inpainted $\mathbf{X}$, brush-box submask selection probability $P$
1: Initialize mask $\mathbf{M}$ with the same size of $\mathbf{X}$
2: Generate a random float $R$ between 0 and 1
3: **if** $R < P$ **then**
4:     Draw $n$ irregular submasks, where $n$ is a random integer drawn from a uniform distribution of a specified range.
5:     **for** $i \leftarrow 0$ to $n$ **do**
6:         Select a random starting point $(x, y)$ in the image
7:         Select random length l, width w and angle a of the brush-like submask
8:         Calculate the end point of the segment $(x', y')$ based on $x, y, a$, and $l$
9:         Generate an brush-like submask in $\mathbf{M}$ from $(x, y)$ to $(x', y')$ with brush width $w$
10:         $x, y \leftarrow x', y'$
11:     **end for**
12: **else**
13:     Draw $n$ irregular submasks, where $n$ is a random integer drawn from a uniform distribution of a specified range.
14:     **for** $i \leftarrow 0$ to $n$ **do**
15:         Select a random size $(h, w)$ and position $(x, y)$ of the submask
16:         Generate a box-like submask based on the selected size and position
17:     **end for**
18: **end if**
19: **return** the generated mask $\mathbf{M}$

---

(*a.k.a.*, the *second inpainting network*). We generate a *second inpainted image* (*a.k.a.*, the *re-inpainted image*) $\hat{\mathbf{X}}_2 = F_2(\hat{\mathbf{X}}_1 \odot \mathbf{M}_2, \mathbf{M}_2)$.

In our proposed evaluation framework, we start with an original image $\mathbf{X}$ masked with a normal mask $\mathbf{M}_1$, which is commonly encountered in real-world applications. The inpainting methods under testing are then applied to inpaint the first masked image $\mathbf{X} \odot \mathbf{M}_1$, resulting in a first inpainted image $\hat{\mathbf{X}}_1$. Subsequently, we apply multiple patch masks $\mathbf{M}_2$ to the first inpainted image and use a chosen inpainting network $F_2(\cdot)$ to further inpaint it, generating a set of inpainted images $\{\hat{\mathbf{X}}_2^k |_{k=1}^K\}$. We empirically choose K as 10, and the results are collectively aggregated.

To ensure unbiased evaluations and avoid style similarities between the first and second inpainting networks, we employ a selective masking approach. Specifically, only the parts of the first inpainted image that have not been previously masked are masked again. In other words, after collecting the patch mask $\mathbf{M}_p$, we first preprocess it to obtain $\mathbf{M}_2 = 1 - (1 - \mathbf{M}_p) \odot \mathbf{M}_1$, then we mask $\hat{\mathbf{X}}_1$ with $\mathbf{M}_2$. Our proposed consistency metric for evaluating image inpainting methods can be formulated as:

$$D(F_1) = \frac{1}{K} \sum_{i=1}^{K} d(\hat{\mathbf{X}}_1, \hat{\mathbf{X}}_2^i), \tag{1}$$

here, the sub-metric $d(\cdot, \cdot)$, which can be based on common metrics like PSNR, SSIM (Wang et al., 2004), and LPIPS (Zhang et al., 2018), is employed to compare the first inpainted image $\hat{\mathbf{X}}_1$ with each second inpainted image $\hat{\mathbf{X}}_2^i$. These second inpainted images are generated using the inpainting method $F_2(\cdot)$ and the patch-wise mask $\mathbf{M}_2$. The resulting sub-metric values are then averaged over $K$ iterations to obtain the final metric value $D(F_1)$. This metric quantifies the consistency between the first inpainted images and the second inpainted images, providing an objective measure for the multi-pass self-consistency of the images produced by the inpainting methods.

### 3.3 ALLEVIATING BIAS WITH PATCH MASKS

Most existing evaluation metrics for image inpainting involve direct comparisons between the original and the restored images, either in the pixel space or the embedded feature space. However, metrics such as Mean Squared Error (MSE), Peak Signal-to-Noise Ratio (PSNR), Structural Similarity Index (SSIM) (Wang et al., 2004), and Learned Perceptual Image Patch Similarity (LPIPS)

---

**Algorithm 2** Patch Mask Generator

---

**Require:** The image to be masked $\mathbf{X}$, size of each patch $S$, ratio of the masked region $P$
 1: Initialize mask $\mathbf{M}$ with the same size of $\mathbf{X}$
 2: **for** each patch of size $S$ in $\mathbf{M}$ **do**
 3:    Generate a random float $R$ between 0 and 1
 4:    **if** $R \leq P$ **then**
 5:       Set all pixels in the current patch of the $\mathbf{M}$ to 1 (indicating it is masked)
 6:    **else**
 7:       Set all pixels in the current patch of the $\mathbf{M}$ to 0 (indicating it is not masked)
 8:    **end if**
 9: **end for**
10: **return** the generated mask $\mathbf{M}$

---

(Zhang et al., 2018) have limitations. These metrics impose constraints on the feasible solutions, leading to biases toward certain distributions and restricting the diversity of inpainted results.

Algorithm 1 and Algorithm 2 provide detailed descriptions of the commonly used normal mask (Suvorov et al., 2022) in image inpainting tasks and our proposed patch mask. The normal mask obscures connected regions that resemble brush-like or box-like shapes, while the patch mask independently determines whether to mask each patch, resulting in isolated small regions of blocked images. Inpainted images masked by commonly used normal masks in image inpainting tasks exhibit significant variance and can deviate substantially from the original image. As shown in Figure 1 and Figure 3c, normal masks can introduce diverse results in inpainted images. Consequently, similarity-based metrics such as PSNR, LPIPS, and SSIM fail to provide reliable assessments.

The use of patch masks ensures the stability (low variance) of the high-level aspects, while the focus is directed toward the restoration of the low-level details. As a result, the inpainted images exhibit low variance and closely resemble the original image. Figures 3c and 3e showcase examples of inpainted images under normal mask and patch mask conditions, respectively. It is worth noting that the presence of large connected corrupted regions in randomly masked images often leads to the generation of objects that do not exist in the original image.

To further investigate this matter, we present Figure 3f, which offers a comprehensive analysis of the distribution of LPIPS scores among 100 images inpainted using StableDiffusion, employing the same original image and the first mask. The results reveal a notably lower variance in LPIPS scores when patch masking is utilized in comparison to normal masking, thereby indicating the enhanced stability of our proposed metric for evaluation. This figure also highlights that the use of normal masks introduces a high variance in the inpainted images, emphasizing the potential bias introduced when evaluating inpainting methods with unmasked images.

## 4 EXPERIMENTS

In this section, we provide a comprehensive overview of our proposed benchmark for evaluating image inpainting. We begin by presenting the key features and components of the benchmark, highlighting its multi-pass nature, self-consistency, and metric-driven evaluation. Subsequently, we conduct ablative studies to identify the optimal configuration of the benchmark, ensuring its effectiveness in assessing image inpainting methods. Finally, we utilize the selected benchmark setting to compare it with other metrics and evaluate a variety of image inpainting techniques.

In the Appendix, we include detailed quantitative results obtained from our proposed benchmark, as well as the images used for evaluation and the code implementation of our benchmark.

### 4.1 IMPLEMENTATION DETAILS

**Inpainting Methods and Dataset** We evaluate the inpainting methods $F_1$ performance of five methods: DeepFillv2 (Yu et al., 2019), EdgeConnect (Nazeri et al., 2019), CoModGAN (Zhao et al., 2021), StableDiffusion (Rombach et al., 2022), and LaMa (Suvorov et al., 2022), using a dataset of 100 images selected from the Places2 dataset (Zhou et al., 2017) with resolution $512 \times 512$. These

methods are chosen to represent a diverse range of state-of-the-art inpainting techniques. We use $K = 10$ different patch masks in Eqn. 1. In Eqn. 1, we use LPIPS Zhang et al. (2018) for the sub-metric $d(\cdot, \cdot)$. Please refer to Section A.1 for analyses of other sub-metric choices.

**Masks**  To assess the performance of the inpainting methods, we employ different types of masks. For the original images $\mathbf{X}$, a normal mask $\mathbf{M}_1$ is applied, while for the first inpainted images $\hat{\mathbf{X}}_1$, a patch mask $\mathbf{M}_2$ is utilized. The first mask ratio is varied within the ranges of 0-20%, 20%-40%, and 40%-60%. A higher ratio indicates a more challenging task of recovering the damaged regions. The second mask ratio is fixed at 20%, 40%, and 60% to provide concordance in the evaluation. To generate random masks within the specified ranges or patch masks with the specified ratio, we utilize the method described in Algorithm 1 and Algorithm 2.

## 4.2 CHOICE OF METRIC OBJECTIVE

In Eqn. 1, we discussed the use of the evaluation between the first inpainted image $\hat{\mathbf{X}}_1$ and the second inpainted images $\hat{\mathbf{X}}_2$ as the final consistency metric for image inpainting methods. In this section, we explore different options for this objective and present the rationale behind our choice. We evaluate three different metrics in Table 1 with a fixed second mask ratio of 40%:

Table 1: Quantitative results obtained using StableDiffusion as the second inpainting network with a fixed second mask ratio of 40%.

| | | Metric Objective | | |
|---|---|---|---|---|
| First Mask Ratio | Method | 1-to-0 | 2-to-0 | 2-to-1 |
| 0%-20% | DeepFillv2 | 0.0586 | 0.3183 | 0.2860 |
| | EdgeConnect | 0.0649 | 0.3254 | 0.2910 |
| | CoModGAN | 0.0590 | 0.3177 | 0.2823 |
| | StableDiffusion | 0.0555 | 0.3139 | **0.2758** |
| | LaMa | **0.0491** | **0.3093** | 0.2817 |
| 20%-40% | DeepFillv2 | 0.1714 | 0.3705 | 0.2635 |
| | EdgeConnect | 0.1832 | 0.3832 | 0.2790 |
| | CoModGAN | 0.1683 | 0.3654 | 0.2552 |
| | StableDiffusion | 0.1650 | 0.3608 | **0.2384** |
| | LaMa | **0.1464** | **0.3464** | 0.2581 |
| 40%-60% | DeepFillv2 | 0.2735 | 0.4288 | 0.2435 |
| | EdgeConnect | 0.2859 | 0.4394 | 0.2668 |
| | CoModGAN | 0.2620 | 0.4148 | 0.2326 |
| | StableDiffusion | 0.2643 | 0.4144 | **0.2089** |
| | LaMa | **0.2352** | **0.3909** | 0.2415 |

Table 2: Statistics of the proposed metric for various combinations of first and second mask ratios.

| | | Second Mask Ratio | | |
|---|---|---|---|---|
| First Mask Ratio | Method | 20% | 40% | 60% |
| 0%-20% | DeepFillv2 | 0.2189 | 0.2860 | 0.3471 |
| | EdgeConnect | 0.2231 | 0.2910 | 0.3540 |
| | CoModGAN | 0.2161 | 0.2823 | 0.3433 |
| | StableDiffusion | **0.2101** | **0.2758** | **0.3359** |
| | LaMa | 0.2161 | 0.2817 | 0.3416 |
| 20%-40% | DeepFillv2 | 0.2113 | 0.2635 | 0.3100 |
| | EdgeConnect | 0.2252 | 0.2790 | 0.3274 |
| | CoModGAN | 0.2037 | 0.2552 | 0.3015 |
| | StableDiffusion | **0.1874** | **0.2384** | **0.2835** |
| | LaMa | 0.2071 | 0.2581 | 0.3028 |
| 40%-60% | DeepFillv2 | 0.2026 | 0.2435 | 0.2789 |
| | EdgeConnect | 0.2258 | 0.2668 | 0.3051 |
| | CoModGAN | 0.1926 | 0.2326 | 0.2678 |
| | StableDiffusion | **0.1702** | **0.2089** | **0.2429** |
| | LaMa | 0.2025 | 0.2415 | 0.2759 |

- **Original-First**: This metric utilizes a sub-metric that compares the original image $\mathbf{X}$ with the first inpainted image $\hat{\mathbf{X}}_1$. This approach is commonly used for conventional evaluation in image inpainting. However, as previously mentioned, this metric can introduce biases in the evaluation process.

- **Original-Second**: This metric employs a sub-metric that compares the original image $\mathbf{X}$ with the second inpainted image $\hat{\mathbf{X}}_2$. As shown in Table 1, the results of **Original-Second** exhibit a similar tendency to **Original-First**, indicating the persistence of biases in this metric.

- **First-Second**: This metric employs a sub-metric that compares the first inpainted image $\hat{\mathbf{X}}_1$ with the second inpainted image $\hat{\mathbf{X}}_2$, without involving the original image $\mathbf{X}$. As mentioned earlier, this metric captures the self-consistency of the inpainting method. The results differ significantly from those of **Original-First** and **Original-Second**.

Considering that **First-Second** is the only metric objective that does not rely on the original image $\mathbf{X}$, we select it as the metric objective for our proposed benchmark. By focusing on the similarity between the first and second inpainted images, we aim to capture the self-consistency of the inpainted images and provide a reliable and unbiased assessment of the inpainting performance. This metric choice aligns with our goal of evaluating the ability of inpainting methods to maintain context consistency.

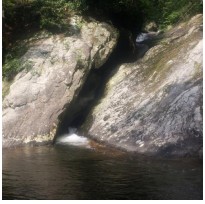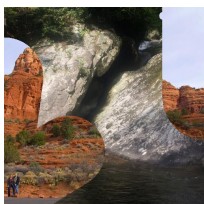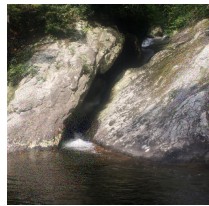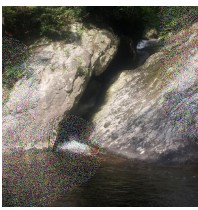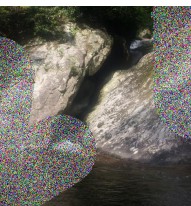

Figure 4: Examples of synthesized images, from left to right: natural image, blended image, noised image with $\sigma$=0.1, noised image with $\sigma$=0.3 and noised image with $\sigma$=1.0.

## 4.3 CHOICE OF SECOND MASK RATIO

Table 2 illustrates the variation of the second mask ratio to examine the consistency of the proposed evaluation metric. As previously mentioned in the subsections, we adopt **First-Second** as the objective metric, employ LPIPS as the sub-metric, and utilize StableDiffusion as the second inpainting network. Additionally, we vary the first mask ratio to assess the consistency of our findings.

From the table, it is evident that our proposed method demonstrates stability across different second mask ratios.

## 4.4 VALIDATION ON SYNTHESIZED INPAINTING IMAGES

To intuitively demonstrate the capabilities of our framework in evaluating inpainted images, we have synthesized several categories of bad inpainting results. We compute the scores for both the synthesized images and the natural images using our approach and subsequently compare these scores. In more detail, we employ our subset of 100 inpainted images $\{\mathbf{X}_1\}$ from

Table 3: Statistics of the proposed metric on synthesized images.

| Processing Method | First Mask Ratio | | |
|---|---|---|---|
| | 0%-20% | 20%-40% | 40%-60% |
| Natural | **0.2778** | **0.2455** | **0.2206** |
| Blend | 0.2794 | 0.2484 | 0.2279 |
| Noise 0.1 | 0.3015 | 0.3034 | 0.3044 |
| Noise 0.3 | 0.3060 | 0.3210 | 0.3341 |
| Noise 1.0 | 0.3085 | 0.3281 | 0.3452 |

Places2 dataset and the corresponding 100 random masks $\{\mathbf{M}_1\}$ for our experiments. In the first setting, we aim to emulate inpainting results that maintain local consistency in most areas yet lack global content consistency. To achieve this, we choose a distinct random image, denoted as $\mathbf{I}$, from the set $\{\mathbf{X}_1\}$ to populate the masked region of our original image $\mathbf{X}$. Given that the random mask associated with $\mathbf{X}$ is $\mathbf{M}_1$, the inpainted image $\hat{\mathbf{X}}_1$ is formulated as:

$$\hat{\mathbf{X}}_1 = \mathbf{X} \odot \mathbf{M}_1 + \mathbf{I} \odot (1 - M_1) \tag{2}$$

In the second setting, we introduce Gaussian noise with varying magnitudes to the masked region in order to simulate inpainting results that may lack detail and fidelity. This can be mathematically represented as:

$$\hat{\mathbf{X}}_1 = \mathbf{X} \odot \mathbf{M}_1 + (\mathbf{X} + \mathcal{N}(0, \sigma^2)) \odot (1 - \mathbf{M}_1) \tag{3}$$

We empirically select three distinct magnitudes of Gaussian noise. The first simulates subtle noise, allowing details within the noisy region to remain discernible. The second introduces moderate noise, preserving only the broader structure of the affected area. The third applies intense noise, making the noisy region nearly indistinguishable. These scenarios correspond to values of $\sigma$ being 0.1, 0.3, and 1.0, respectively. The subsequent stages of our experiment follow our framework detailed in 3.2, we apply multiple patch masks with a ratio of 40% then inpaint them using Stable Diffusion, the sub-metric $d(\cdot, \cdot)$ is set to LPIPS only.

We present examples of the synthesized images in Figure 4. Upon reviewing the figure, it becomes evident that these synthesized images exhibit lower quality in comparison to natural images. The content of blended images lacks consistency, while the noise-infused images demonstrate blurred inappropriate outcomes. As Table 3 shows, all categories of synthesized poorly inpainting images

yield larger values of Eq. 1, which validates the effectiveness of our approach intuitively: our proposed approach can both evaluate the appropriateness and fidelity of inpainted images.

## 4.5 OVERALL EVALUATION OF THE FIRST INPAINTING NETWORK

Table 4: Quantitative results of two NR-IQA metrics, namely MUSIQ and PAR, along with our proposed metric and human evaluations.

|  |  | Method | MUSIQ | PAR(%) | **Ours** | Human(%) |
|---|---|---|---|---|---|---|
| | | | | | **Metrics** | |
| **First Mask Ratio** | 0% – 20% | DeepFillv2 | 64.62 | **72.60** | 0.2859 | 8.72 |
| | | EdgeConnect | 64.89 | 81.39 | 0.2911 | 5.39 |
| | | CoModGAN | 65.85 | 83.30 | 0.2823 | 16.91 |
| | | StableDiffusion | **65.86** | 87.58 | **0.2760** | **45.53** |
| | | LaMa | 65.61 | 74.42 | 0.2815 | 23.45 |
| | 20% – 40% | DeepFillv2 | 61.53 | **24.38** | 0.2634 | 1.23 |
| | | EdgeConnect | 62.74 | 35.04 | 0.2789 | 1.39 |
| | | CoModGAN | 65.24 | 33.48 | 0.2552 | 20.67 |
| | | StableDiffusion | **65.73** | 36.72 | **0.2382** | **58.03** |
| | | LaMa | 63.94 | 30.10 | 0.2581 | 18.68 |
| | 40% – 60% | DeepFillv2 | 58.96 | **16.35** | 0.2432 | 0.60 |
| | | EdgeConnect | 61.19 | 26.99 | 0.2670 | 0.21 |
| | | CoModGAN | 64.96 | 23.55 | 0.2325 | 27.61 |
| | | StableDiffusion | **65.07** | 26.88 | **0.2089** | **59.39** |
| | | LaMa | 62.18 | 23.56 | 0.2418 | 12.19 |

In this section, we provide a comprehensive evaluation of the first inpainting network based on the established settings from the previous subsections. The objective metric **First-Second** is employed, with LPIPS as the sub-metric. We select StableDiffusion as the second inpainting network and set the second mask ratio to 40%. To benchmark our proposed method, we compare it with two No-Reference Image Quality Assessment (NR-IQA) metrics, MUSIQ (Ke et al., 2021) and PAR (Zhang et al., 2022), as well as a user study conducted by 100 professional human evaluators. The user study scores are determined by assessing the most plausible images among all the inpainted images generated by different inpainting methods. The results are summarized in Table 4.

From the human evaluation results, we observe that StableDiffusion emerges as the top-performing method. While the advantages of StableDiffusion may not be evident when the first mask ratio is low, as all methods can easily restore small damaged areas, its superiority becomes apparent as the first mask ratio increases. This can be attributed to its extensive training dataset and advanced model structure. The results of PAR, however, differ significantly from human evaluation. Conversely, both MUSIQ and our proposed benchmark closely align with the conclusions of human evaluation, indicating their effectiveness. In comparison to MUSIQ, our proposed method offers the advantage of not requiring training with image quality annotations, thereby providing flexibility and cost-effectiveness.

## 5 CONCLUSIONS

In this paper, we introduce a novel evaluation framework that harnesses the capabilities of aggregated multi-pass image inpainting. Our proposed self-supervised metric achieves remarkable performance in both scenarios with or without access to unmasked images. Instead of relying solely on similarity to the original images in terms of pixel space or feature space, our method emphasizes intrinsic self-consistency. This approach enables the exploration of diverse and viable inpainting solutions while mitigating biases. Through extensive experimentation across various baselines, we establish the strong alignment between our method and human perception, which is further corroborated by a comprehensive user study.

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

# A APPENDIX

In this section, we further explore the details of our experiment, presenting the comprehensive quantitative results of our proposed benchmark, along with some examples from the second inpainting network. The code for our proposed benchmark is available on Google Drive; please refer to the provided URL [1].

## A.1 CHOICE OF SUB-METRIC AND THE SECOND INPAINTING NETWORK

Table 5: Quantitative results showing the impact of varying the first mask ratio and second inpainting networks.

| | | | First Mask 0%-20% | | | First Mask 20%-40% | | | First Mask 40%-60% | | |
|---|---|---|---|---|---|---|---|---|---|---|---|
| | | Method | PSNR | SSIM | LPIPS | PSNR | SSIM | LPIPS | PSNR | SSIM | LPIPS |
| Second Inpainting Methods | StableDiffusion | DeepFillv2 | 21.7949 | 0.6487 | 0.2860 | 22.8094 | 0.6855 | 0.2635 | 23.7716 | 0.7249 | 0.2435 |
| | | EdgeConnect | **21.8444** | 0.6498 | 0.2910 | 22.7964 | 0.6771 | 0.2790 | 23.6027 | 0.7021 | 0.2668 |
| | | CoModGAN | 21.7173 | 0.6465 | 0.2823 | 22.4921 | 0.6773 | 0.2552 | 23.2653 | 0.7080 | 0.2326 |
| | | StableDiffusion | 21.8031 | **0.6586** | **0.2758** | 22.7357 | **0.7053** | **0.2384** | 23.4685 | **0.7431** | **0.2089** |
| | | LaMa | 21.8414 | 0.6507 | 0.2817 | **22.8644** | 0.6855 | 0.2581 | **23.8487** | 0.7174 | 0.2415 |
| | LaMa | DeepFillv2 | **26.0877** | **0.8804** | 0.1335 | **28.4204** | **0.9142** | 0.1050 | 28.6469 | 0.9278 | 0.0867 |
| | | EdgeConnect | 26.0820 | 0.8803 | 0.1330 | 27.4104 | 0.9077 | 0.1052 | 28.6063 | 0.9273 | 0.0837 |
| | | CoModGAN | 26.0248 | 0.8797 | 0.1322 | 27.3358 | 0.9072 | 0.1043 | 28.5275 | 0.9269 | 0.0833 |
| | | StableDiffusion | 26.0613 | 0.8798 | **0.1319** | 27.3632 | 0.9069 | **0.1040** | 28.5544 | 0.9265 | **0.0822** |
| | | LaMa | 26.0836 | 0.8804 | 0.1321 | 28.4181 | 0.9129 | 0.1042 | **28.6547** | **0.9279** | 0.0833 |
| | DeepFillv2 | DeepFillv2 | **24.8895** | **0.8614** | 0.1583 | **26.2330** | **0.8936** | 0.1278 | 27.4044 | **0.9158** | 0.1041 |
| | | EdgeConnect | 24.8560 | 0.8612 | 0.1573 | 26.1859 | 0.8926 | 0.1257 | **27.4083** | 0.9157 | 0.1000 |
| | | CoModGAN | 24.8108 | 0.8605 | 0.1565 | 26.1428 | 0.8923 | 0.1244 | 27.3103 | 0.9149 | 0.0994 |
| | | StableDiffusion | 24.8407 | 0.8605 | **0.1564** | 26.1738 | 0.8923 | **0.1234** | 27.3663 | 0.9150 | **0.0981** |
| | | LaMa | 24.8616 | 0.8612 | 0.1567 | 26.1659 | 0.8929 | 0.1251 | 27.3760 | 0.9158 | 0.1003 |

In Eqn. 1, we have three different choices for the sub-metric $d(\cdot, \cdot)$:

- PSNR (Peak Signal-to-Noise Ratio): PSNR is a commonly used objective metric for image quality assessment. It measures the ratio between the maximum possible power of a signal and the power of the noise present in the signal.

- SSIM (Wang et al., 2004) (Structural Similarity Index): SSIM is another widely used metric for evaluating the perceptual quality of images. It measures the structural similarity between the original and distorted images, taking into account their luminance, contrast, and structural information.

- LPIPS (Zhang et al., 2018) (Learned Perceptual Image Patch Similarity): LPIPS is a metric that utilizes deep neural networks to measure the perceptual similarity between images. Unlike PSNR and SSIM, which rely on handcrafted features, LPIPS learns feature representations from large-scale image datasets.

Regarding the second inpainting network, denoted as $F_2$, we alternate between StableDiffusion, DeepFillv2, and LaMa. This selection ensures consistent evaluation results across different choices of the second inpainting method.

In Table 5, we vary the first mask ratio, all three sub-metrics, and the second inpainting networks while keeping the second mask ratio fixed. From the results, we observe an interesting phenomenon: the choice of the second inpainting network impacts the results of PSNR and SSIM. Specifically, if we use DeepFillv2 as the second inpainting network, DeepFillv2 yields the best results in terms of PSNR and SSIM. Conversely, if we switch the second inpainting network to LaMa, LaMa becomes the best first inpainting network. This suggests that the generated results from the second network tend to exhibit a similar style to those from the first network when the same model is used for both. However, when different models are employed, there may be a variance in image style, which in turn leads to a decline in the metrics that are based on pixel-level features, rather than on learned perceptual features.

---

[1] https://drive.google.com/drive/folders/1NgYy8gUsGNaNwcuBfNVzi6LL30XxJwBO

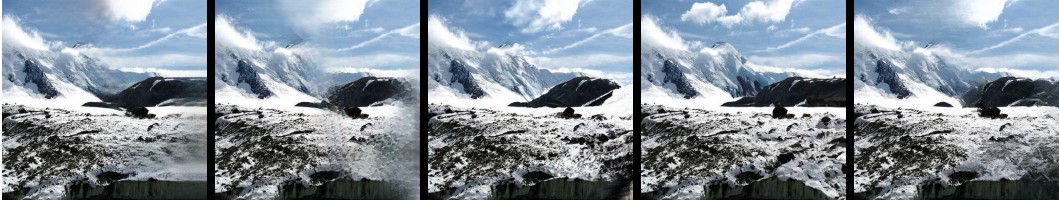

Figure 5: The arrangement of inpainted images shown to participants, from left to right: DeepFillv2, EdgeConnect, CoModGAN, StableDiffusion, and LaMa

On the other hand, we found that LPIPS remains consistent across different second inpainting networks. This can be attributed to the fact that LPIPS is based on perceptual evaluation. Therefore, we chose LPIPS as the sub-metric in our evaluation to ensure consistent and reliable results.

## A.2 EXAMPLE INPAINTED IMAGES FROM THE SECOND INPAINTING NETWORK

In Figure 6, we present an example of inpainted images from the second inpainting network. We select the first mask ratio in the interval of 20-40%. We then show 5 different second masks with a mask ratio of 40%, along with the corresponding inpainted results for different first inpainting methods. From the figure, we can observe varying degrees of self-consistency among the inpainted images produced by different first inpainting methods. For other settings of our benchmark, please refer to the provided code.

## A.3 DATASET AND EXPERIMENT DETAILS

We randomly select 100 $512 \times 512$ images from Places2 to form our dataset, which can be accessed at https://drive.google.com/drive/folders/1NgYy8gUsGNaNwcuBfNVzi6LL30XxJwBO?usp=sharing. To further validate the comprehensiveness of our chosen subset, we expanded our evaluation to include an additional 10 and 1000 images from the Places2 dataset, applying our framework to each set. We set the first mask ratio ranging from 20% to 40% and the second mask ratio 40%. StableDiffusion is employed as both the first and second inpainting network. As illustrated in Figure 7a, the score distributions derived from our framework remain stable across datasets of different sizes, which demonstrates the representativeness of our dataset.

The choice of the number of second masks per first inpainted image is a problem of balancing between computing efficiency and measurement stability. While a greater number of patch masks would provide a more stable and unbiased result, it would also increase the computation time. We empirically choose 10 masks to get the proper balance, ensuring both stable results and acceptable computational requirements. As shown in Figure 7b, we conducted experiments with K=10, 100 and 1000 to a single first inpainted image. The second mask ratio is set to 40% and we employed StableDiffusion as the second inpainting network. For the overall evaluation of the first inpainting networks, our framework is initialized with three different random seeds, and we report the average score in Table 4. The standard deviation for each case remains within 0.0003.

## A.4 DETAILS ON HUMAN EVALUATION

We applied inpainting for each randomly masked image using five different methods: DeepFillv2, EdgeConnect, CoModGAN, StableDiffusion, and LaMa. The inpainted images were arranged in a row without any text descriptions, as shown in Figure 5. We then surveyed 100 unpaid volunteers, all from computer science or related disciplines. Each participant was given 100 rows of these inpainted images to evaluate. They were instructed: "For each row, you'll see images inpainted by five different methods from the same original image. Please select the one that appears the most visually natural and contextually consistent to you." The human evaluation score is defined as the average percentage of times a particular method was chosen as producing the best inpainting result.

### A.5 LIMITATIONS & SOCIETAL IMPACT

**Limitations**   While our framework allows for more diversified inpainting results, the per-image evaluation time is slower. In comparison to the direct LPIPS measurement, our method incorporates an additional inpainting network. The per image per second mask computation time is 1x to 10x times slower than direct LPIPS, depending on the second inpainting network used. As an example, reproducing Table 2 with K=10 would require 45 hours on a single A5000 GPU.

**Societal Impact**   Development in general visual generative models including image inpainting models is a double-edged sword. On the one hand, these models open up various new applications and creative workflows. For instance, image inpainting can be used as a procedure in digital drawing, which may effectively boost the efficiency of digital artists. On the other hand, such models can be misused to produce and distribute altered data, potentially leading to misinformation and spam. Thus, it's crucial to keep the deployment of such models under proper usage and regulation.

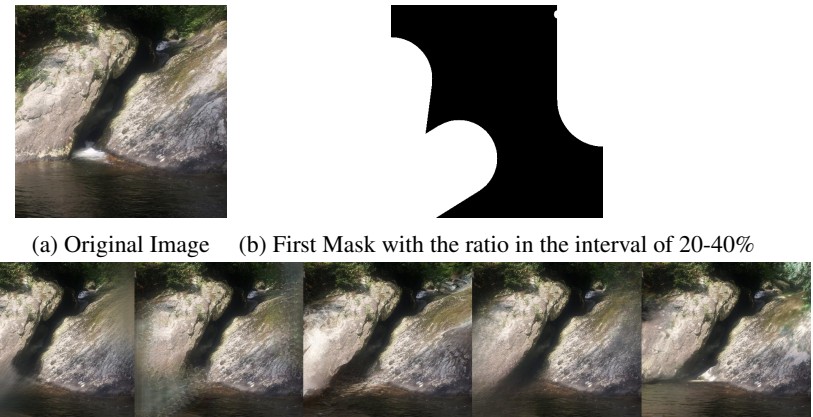

(a) Original Image    (b) First Mask with the ratio in the interval of 20-40%

(c) First Inpainted Images, from left to right: DeepFillv2, EdgeConnect, CoModGAN, LaMa, and StableDiffusion

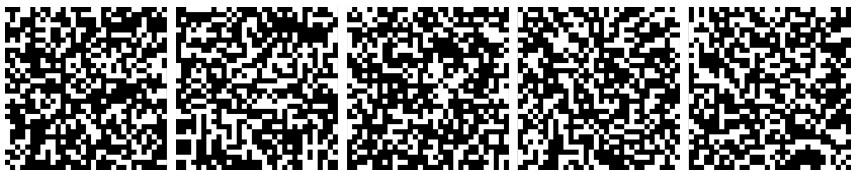

(d) Second Masks with ratio 40%

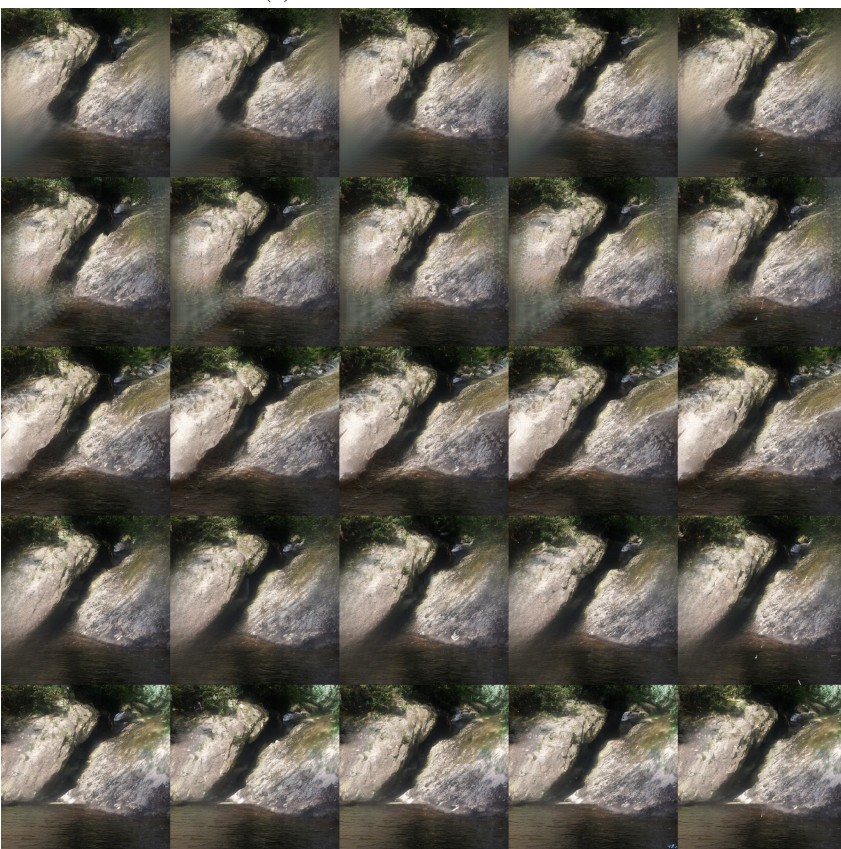

(e) Second Inpainted Images: Each row represents the results obtained from different first inpainting methods, namely DeepFillv2, EdgeConnect, CoModGAN, LaMa, and StableDiffusion. Each column corresponds to a different second inpainting mask.

Figure 6: Example masks and inpainted images.

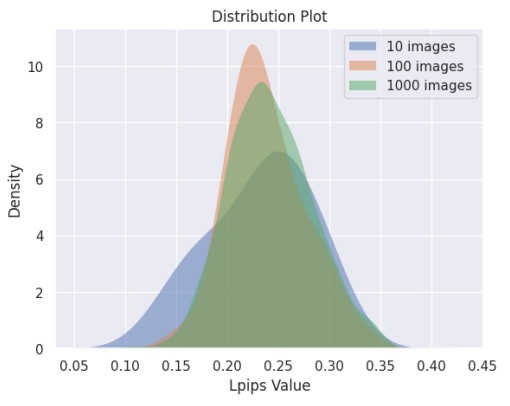
(a) Comparison of datasets of different sizes

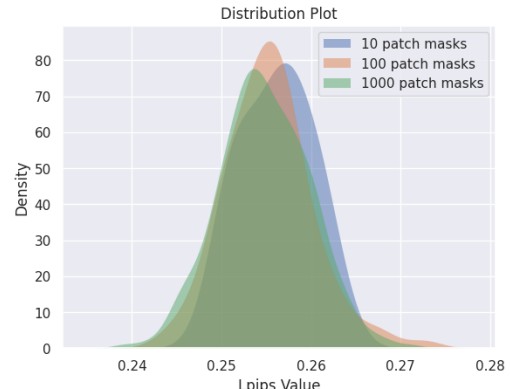
(b) Comparison of patch mask numbers

## A.6 FULL QUANTITATIVE RESULTS

In Section 4, we conducted several ablative studies of our proposed benchmark. Here, we present the complete results of our benchmark, evaluating different inpainting methods. We evaluate the performance of the inpainting methods $F_1$ using five techniques: DeepFillv2 Yu et al. (2019), EdgeConnect Nazeri et al. (2019), CoModGAN Zhao et al. (2021), StableDiffusion Rombach et al. (2022), and LaMa Suvorov et al. (2022). These methods are chosen to represent a diverse range of state-of-the-art inpainting techniques. We use $K = 10$ different patch masks in Eqn. 1. To assess the performance of the inpainting methods, we employ different types of masks. For the original images $\mathbf{X}$, a normal mask $\mathbf{M}_1$ is applied, while for the first inpainted images $\hat{\mathbf{X}}_1$, a patch mask $\mathbf{M}_2$ is utilized. The first mask ratio is varied within the ranges of 0-20%, 20%-40%, and 40%-60%. A higher ratio indicates a more challenging task of recovering the damaged regions. The second mask ratio is fixed at 20%, 40%, and 60% to ensure consistency in the evaluation. To generate random masks within the specified ranges or generate patch masks with the specified ratio, we utilize the methods described in Algorithm 1 and Algorithm 2. We vary the metric objective among **Original-First**, **Original-Second**, and **First-Second**, and vary the sub-metric to include PSNR, SSIM, and LPIPS. The results can be found in Tables 6-14. It is important to note that the results of **Original-First** remain identical across different second inpainting methods. These results provide further support for the conclusions made in Section 4.

Table 6: Quantitative results on a subset of the Places2 dataset, with varying first mask ratios ranging from 0% to 20%, and a fixed second mask ratio of 20%.

| | | | Original-First Inpainting Metrics | | | Original-Second Inpainting Metrics | | | First-Second Inpainting Metrics | | |
|---|---|---|---|---|---|---|---|---|---|---|---|
| | | Method | PSNR | SSIM | LPIPS | PSNR | SSIM | LPIPS | PSNR | SSIM | LPIPS |
| Second Inpainting Methods | StableDiffusion | DeepFillv2 | 28.1927 | 0.9429 | 0.0586 | 21.8288 | 0.6806 | 0.2532 | 23.8474 | 0.7110 | 0.2189 |
| | | EdgeConnect | 27.0888 | 0.9404 | 0.0649 | 21.5279 | 0.6780 | 0.2597 | **23.8937** | 0.7119 | 0.2231 |
| | | CoModGAN | 27.1559 | 0.9367 | 0.059 | 21.3926 | 0.6777 | 0.2535 | 23.7084 | 0.7100 | 0.2161 |
| | | StableDiffusion | 27.0113 | 0.9369 | 0.0555 | 21.2203 | 0.6747 | 0.2503 | 23.8512 | **0.7217** | **0.2101** |
| | | LaMa | **29.3233** | **0.9481** | **0.0491** | **22.1120** | **0.6854** | **0.2450** | 23.8624 | 0.7130 | 0.2161 |
| | LaMa | DeepFillv2 | 28.1927 | 0.9429 | 0.0586 | 24.8951 | 0.8875 | 0.1237 | **30.0454** | 0.9446 | 0.0670 |
| | | EdgeConnect | 27.0888 | 0.9404 | 0.0649 | 24.3749 | 0.8850 | 0.1295 | 30.0428 | 0.9446 | 0.0666 |
| | | CoModGAN | 27.1559 | 0.9367 | 0.0590 | 24.1829 | 0.8812 | 0.1236 | 29.9844 | 0.9443 | 0.0662 |
| | | StableDiffusion | 27.0113 | 0.9369 | 0.0555 | 24.0408 | 0.8814 | 0.1200 | 30.0221 | 0.9444 | **0.0661** |
| | | LaMa | **29.3233** | **0.9481** | **0.0491** | **25.4690** | **0.8928** | **0.1140** | 30.0443 | **0.9447** | 0.0662 |
| | DeepFillv2 | DeepFillv2 | 28.1927 | 0.9429 | 0.0586 | 24.4023 | 0.8784 | 0.1349 | **28.8577** | **0.9355** | 0.0787 |
| | | EdgeConnect | 27.0888 | 0.9404 | 0.0649 | 23.9202 | 0.8756 | 0.1412 | 28.8462 | 0.9352 | 0.0787 |
| | | CoModGAN | 27.1559 | 0.9367 | 0.0590 | 23.7362 | 0.8722 | 0.1344 | 28.8078 | 0.9355 | 0.0775 |
| | | StableDiffusion | 27.0113 | 0.9369 | 0.0555 | 23.5697 | 0.8723 | 0.1314 | 28.8070 | 0.9353 | **0.0775** |
| | | LaMa | **29.3233** | **0.9481** | **0.0491** | **24.8964** | **0.8836** | **0.1254** | 28.8335 | 0.9355 | 0.0781 |

Table 7: Quantitative results on a subset of the Places2 dataset, with varying first mask ratios ranging from 0% to 20%, and a fixed second mask ratio of 40%.

| | | | Original-First Inpainting Metrics | | | Original-Second Inpainting Metrics | | | First-Second Inpainting Metrics | | |
|---|---|---|---|---|---|---|---|---|---|---|---|
| | | Method | PSNR | SSIM | LPIPS | PSNR | SSIM | LPIPS | PSNR | SSIM | LPIPS |
| Second Inpainting Methods | StableDiffusion | DeepFillv2 | 28.1927 | 0.9429 | 0.0586 | 20.4058 | 0.6195 | 0.3183 | 21.7949 | 0.6487 | 0.2860 |
| | | EdgeConnect | 27.0888 | 0.9404 | 0.0649 | 20.1790 | 0.6169 | 0.3254 | **21.8444** | 0.6498 | 0.2910 |
| | | CoModGAN | 27.1559 | 0.9367 | 0.0590 | 20.1118 | 0.6165 | 0.3177 | 21.7173 | 0.6465 | 0.2823 |
| | | StableDiffusion | 27.0113 | 0.9369 | 0.0555 | 19.9455 | 0.6140 | 0.3139 | 21.8031 | **0.6586** | **0.2758** |
| | | LaMa | **29.3233** | **0.9481** | **0.0491** | **20.6442** | **0.6242** | **0.3093** | 21.8414 | 0.6507 | 0.2817 |
| | LaMa | DeepFillv2 | 28.1927 | 0.9429 | 0.0586 | 23.0158 | 0.8233 | 0.1887 | **26.0877** | **0.8804** | 0.1335 |
| | | EdgeConnect | 27.0888 | 0.9404 | 0.0649 | 22.6460 | 0.8208 | 0.1942 | 26.0820 | 0.8803 | 0.1330 |
| | | CoModGAN | 27.1559 | 0.9367 | 0.0590 | 22.4587 | 0.8168 | 0.1883 | 26.0248 | 0.8797 | 0.1322 |
| | | StableDiffusion | 27.0113 | 0.9369 | 0.0555 | 22.3209 | 0.8169 | 0.1846 | 26.0613 | 0.8798 | **0.1319** |
| | | LaMa | **29.3233** | **0.9481** | **0.0491** | **23.3934** | **0.8286** | **0.1788** | 26.0836 | 0.8804 | 0.1321 |
| | DeepFillv2 | DeepFillv2 | 28.1927 | 0.9429 | 0.0586 | 22.3157 | 0.8043 | 0.2127 | **24.8895** | **0.8614** | 0.1583 |
| | | EdgeConnect | 27.0888 | 0.9404 | 0.0649 | 21.9770 | 0.8017 | 0.2178 | 24.8560 | 0.8612 | 0.1573 |
| | | CoModGAN | 27.1559 | 0.9367 | 0.0590 | 21.8044 | 0.7970 | 0.2121 | 24.8108 | 0.8605 | 0.1565 |
| | | StableDiffusion | 27.0113 | 0.9369 | 0.0555 | 21.6530 | 0.7976 | 0.2087 | 24.8407 | 0.8605 | **0.1564** |
| | | LaMa | **29.3233** | **0.9481** | **0.0491** | **22.6191** | **0.8094** | **0.2028** | 24.8616 | 0.8612 | 0.1567 |

Table 8: Quantitative results on a subset of the Places2 dataset, with varying first mask ratios ranging from 0% to 20%, and a fixed second mask ratio of 60%.

| | | Method | Original-First Inpainting Metrics | | | Original-Second Inpainting Metrics | | | First-Second Inpainting Metrics | | |
|---|---|---|---|---|---|---|---|---|---|---|---|
| | | | PSNR | SSIM | LPIPS | PSNR | SSIM | LPIPS | PSNR | SSIM | LPIPS |
| Second Inpainting Methods | StableDiffusion | DeepFillv2 | 28.1927 | 0.9429 | 0.0586 | 18.8965 | 0.5619 | 0.3784 | 19.8600 | 0.5904 | 0.3471 |
| | | EdgeConnect | 27.0888 | 0.9404 | 0.0649 | 18.7292 | 0.5594 | 0.3870 | 19.9061 | 0.5917 | 0.3540 |
| | | CoModGAN | 27.1559 | 0.9367 | 0.0590 | 18.6641 | 0.5584 | 0.3774 | 19.7730 | 0.5878 | 0.3433 |
| | | StableDiffusion | 27.0113 | 0.9369 | 0.0555 | 18.5568 | 0.5572 | 0.3724 | 19.8725 | **0.6003** | **0.3359** |
| | | LaMa | **29.3233** | **0.9481** | **0.0491** | **19.0951** | **0.5681** | **0.3683** | **19.9228** | 0.5939 | 0.3416 |
| | LaMa | DeepFillv2 | 28.1927 | 0.9429 | 0.0586 | 21.2212 | 0.7434 | 0.2613 | 23.1770 | 0.8005 | 0.2076 |
| | | EdgeConnect | 27.0888 | 0.9404 | 0.0649 | 20.9600 | 0.7409 | 0.2665 | 23.1726 | 0.8003 | 0.2070 |
| | | CoModGAN | 27.1559 | 0.9367 | 0.0590 | 20.7962 | 0.7366 | 0.2604 | 23.1150 | 0.7993 | 0.2056 |
| | | StableDiffusion | 27.0113 | 0.9369 | 0.0555 | 20.6775 | 0.7369 | 0.2565 | 23.1585 | 0.7996 | **0.2052** |
| | | LaMa | **29.3233** | **0.9481** | **0.0491** | **21.4789** | **0.7489** | **0.2510** | **23.1795** | **0.8007** | 0.2052 |
| | DeepFillv2 | DeepFillv2 | 28.1927 | 0.9429 | 0.0586 | 20.3794 | 0.7162 | 0.2973 | **21.9834** | **0.7732** | 0.2446 |
| | | EdgeConnect | 27.0888 | 0.9404 | 0.0649 | 20.1685 | 0.7137 | 0.3025 | 21.9932 | 0.7731 | 0.2439 |
| | | CoModGAN | 27.1559 | 0.9367 | 0.0590 | 20.0005 | 0.7088 | 0.2962 | 21.9093 | 0.7722 | **0.2420** |
| | | StableDiffusion | 27.0113 | 0.9369 | 0.0555 | 19.8807 | 0.7090 | 0.2932 | 21.9453 | 0.7718 | **0.2421** |
| | | LaMa | **29.3233** | **0.9481** | **0.0491** | **20.5805** | **0.7210** | **0.2878** | 21.9731 | 0.7727 | 0.2428 |

Table 9: Quantitative results on a subset of the Places2 dataset, with varying first mask ratios ranging from 20% to 40%, and a fixed second mask ratio of 20%.

| | | Method | Original-First Inpainting Metrics | | | Original-Second Inpainting Metrics | | | First-Second Inpainting Metrics | | |
|---|---|---|---|---|---|---|---|---|---|---|---|
| | | | PSNR | SSIM | LPIPS | PSNR | SSIM | LPIPS | PSNR | SSIM | LPIPS |
| Second Inpainting Methods | StableDiffusion | DeepFillv2 | 20.3649 | 0.8342 | 0.1714 | 18.9643 | 0.6329 | 0.3218 | 24.7064 | 0.7337 | 0.2113 |
| | | EdgeConnect | 19.3181 | 0.8224 | 0.1832 | 18.1145 | 0.6218 | 0.3333 | 24.6340 | 0.7248 | 0.2252 |
| | | CoModGAN | 19.3045 | 0.8164 | 0.1683 | 18.1921 | 0.6179 | 0.3177 | 24.3046 | 0.7267 | 0.2037 |
| | | StableDiffusion | 18.4795 | 0.8092 | 0.1650 | 17.4232 | 0.6079 | 0.3144 | 24.5880 | **0.7551** | **0.1874** |
| | | LaMa | **21.3790** | **0.8444** | **0.1464** | **19.6529** | **0.6419** | **0.2983** | **24.7283** | 0.7334 | 0.2071 |
| | LaMa | DeepFillv2 | 20.3649 | 0.8342 | 0.1714 | 19.9266 | 0.7917 | 0.2216 | 31.3895 | 0.9574 | 0.0538 |
| | | EdgeConnect | 19.3181 | 0.8224 | 0.1832 | 18.9396 | 0.7798 | 0.2324 | 31.3782 | 0.9572 | 0.0527 |
| | | CoModGAN | 19.3045 | 0.8164 | 0.1683 | 18.9256 | 0.7736 | 0.2176 | 31.3002 | 0.9570 | 0.0523 |
| | | StableDiffusion | 18.4795 | 0.8092 | 0.1650 | 18.1397 | 0.7663 | 0.2139 | 31.3187 | 0.9568 | **0.0520** |
| | | LaMa | **21.3790** | **0.8444** | **0.1464** | **20.8076** | **0.8019** | **0.1961** | **31.3897** | **0.9574** | 0.0524 |
| | DeepFillv2 | DeepFillv2 | 20.3649 | 0.8342 | 0.1714 | 19.8145 | 0.7845 | 0.2308 | 30.1645 | 0.9502 | 0.0641 |
| | | EdgeConnect | 19.3181 | 0.8224 | 0.1832 | 18.8420 | 0.7725 | 0.2418 | 30.1266 | 0.9499 | 0.0629 |
| | | CoModGAN | 19.3045 | 0.8164 | 0.1683 | 18.8357 | 0.7663 | 0.2265 | 30.1090 | 0.9499 | 0.0619 |
| | | StableDiffusion | 18.4795 | 0.8092 | 0.1650 | 18.0557 | 0.7593 | 0.2231 | 30.1270 | 0.9498 | **0.0617** |
| | | LaMa | **21.3790** | **0.8444** | **0.1464** | **20.6609** | **0.7947** | **0.2051** | **30.1818** | **0.9502** | 0.0623 |

Table 10: Quantitative results on a subset of the Places2 dataset, with varying first mask ratios ranging from 20% to 40%, and a fixed second mask ratio of 40%.

| | | Method | Original-First Inpainting Metrics | | | Original-Second Inpainting Metrics | | | First-Second Inpainting Metrics | | |
|---|---|---|---|---|---|---|---|---|---|---|---|
| | | | PSNR | SSIM | LPIPS | PSNR | SSIM | LPIPS | PSNR | SSIM | LPIPS |
| Second Inpainting Methods | StableDiffusion | DeepFillv2 | 20.3649 | 0.8342 | 0.1714 | 18.3761 | 0.5868 | 0.3705 | 22.8094 | 0.6855 | 0.2635 |
| | | EdgeConnect | 19.3181 | 0.8224 | 0.1832 | 17.6199 | 0.5765 | 0.3832 | 22.7964 | 0.6771 | 0.2790 |
| | | CoModGAN | 19.3045 | 0.8164 | 0.1683 | 17.7086 | 0.5727 | 0.3654 | 22.4921 | 0.6773 | 0.2552 |
| | | StableDiffusion | 18.4795 | 0.8092 | 0.1650 | 17.0181 | 0.5631 | 0.3608 | 22.7357 | **0.7053** | **0.2384** |
| | | LaMa | **21.3790** | **0.8444** | **0.1464** | **18.9888** | **0.5965** | **0.3464** | **22.8644** | 0.6855 | 0.2581 |
| | LaMa | DeepFillv2 | 20.3649 | 0.8342 | 0.1714 | 19.6776 | 0.7717 | 0.2422 | **28.4204** | **0.9142** | 0.1050 |
| | | EdgeConnect | 19.3181 | 0.8224 | 0.1832 | 18.4914 | 0.7304 | 0.2820 | 27.4104 | 0.9077 | 0.1052 |
| | | CoModGAN | 19.3045 | 0.8164 | 0.1683 | 18.4836 | 0.7240 | 0.2671 | 27.3358 | 0.9072 | 0.1043 |
| | | StableDiffusion | 18.4795 | 0.8092 | 0.1650 | 17.7439 | 0.7166 | 0.2631 | 27.3632 | 0.9069 | **0.1040** |
| | | LaMa | **21.3790** | **0.8444** | **0.1464** | **20.4780** | **0.7804** | **0.2199** | 28.4181 | 0.9129 | 0.1042 |
| | DeepFillv2 | DeepFillv2 | 20.3649 | 0.8342 | 0.1714 | 19.1673 | 0.7281 | 0.2908 | **26.2330** | **0.8936** | 0.1278 |
| | | EdgeConnect | 19.3181 | 0.8224 | 0.1832 | 18.2762 | 0.7153 | 0.3012 | 26.1859 | 0.8926 | 0.1257 |
| | | CoModGAN | 19.3045 | 0.8164 | 0.1683 | 18.2782 | 0.7087 | 0.2861 | 26.1428 | 0.8923 | 0.1244 |
| | | StableDiffusion | 18.4795 | 0.8092 | 0.1650 | 17.5598 | 0.7020 | 0.2819 | 26.1738 | 0.8923 | **0.1234** |
| | | LaMa | **21.3790** | **0.8444** | **0.1464** | **19.8603** | **0.7375** | **0.2652** | 26.1659 | 0.8929 | 0.1251 |

Table 11: Quantitative results on a subset of the Places2 dataset, with varying first mask ratios ranging from 20% to 40%, and a fixed second mask ratio of 60%.

| | | Method | Original-First Inpainting Metrics | | | Original-Second Inpainting Metrics | | | First-Second Inpainting Metrics | | |
|---|---|---|---|---|---|---|---|---|---|---|---|
| | | | PSNR | SSIM | LPIPS | PSNR | SSIM | LPIPS | PSNR | SSIM | LPIPS |
| Second Inpainting Methods | StableDiffusion | DeepFillv2 | 20.3649 | 0.8342 | 0.1714 | 17.5937 | 0.5434 | 0.4152 | 20.9702 | 0.6408 | 0.3100 |
| | | EdgeConnect | 19.3181 | 0.8224 | 0.1832 | 16.9604 | 0.5336 | 0.4289 | 20.9867 | 0.6329 | 0.3274 |
| | | CoModGAN | 19.3045 | 0.8164 | 0.1683 | 17.0404 | 0.5293 | 0.4094 | 20.7521 | 0.6323 | 0.3015 |
| | | StableDiffusion | 18.4795 | 0.8092 | 0.1650 | 16.4499 | 0.5211 | 0.4028 | 20.9876 | **0.6604** | **0.2835** |
| | | LaMa | **21.3790** | **0.8444** | **0.1464** | **18.1273** | **0.5545** | **0.3897** | 21.0522 | 0.6423 | 0.3028 |
| | LaMa | DeepFillv2 | 20.3649 | 0.8342 | 0.1714 | 18.7276 | 0.6817 | 0.3280 | 24.5094 | 0.8472 | 0.1664 |
| | | EdgeConnect | 19.3181 | 0.8224 | 0.1832 | 17.9030 | 0.6694 | 0.3374 | 24.4951 | 0.8466 | 0.1633 |
| | | CoModGAN | 19.3045 | 0.8164 | 0.1683 | 17.9024 | 0.6628 | 0.3222 | 24.4214 | 0.8457 | 0.1617 |
| | | StableDiffusion | 18.4795 | 0.8092 | 0.1650 | 17.2245 | 0.6554 | 0.3176 | 24.4574 | 0.8454 | **0.1613** |
| | | LaMa | **21.3790** | **0.8444** | **0.1464** | **19.3694** | **0.6922** | **0.3010** | 24.5213 | **0.8475** | 0.1613 |
| | DeepFillv2 | DeepFillv2 | 20.3649 | 0.8342 | 0.1714 | 18.3622 | 0.6609 | 0.3559 | **23.3539** | **0.8263** | 0.1962 |
| | | EdgeConnect | 19.3181 | 0.8224 | 0.1832 | 17.5654 | 0.6480 | 0.3655 | 23.2992 | 0.8251 | 0.1932 |
| | | CoModGAN | 19.3045 | 0.8164 | 0.1683 | 17.5695 | 0.6400 | 0.3508 | 23.2127 | 0.8236 | 0.1917 |
| | | StableDiffusion | 18.4795 | 0.8092 | 0.1650 | 16.9239 | 0.6338 | 0.3466 | 23.2890 | 0.8238 | **0.1910** |
| | | LaMa | **21.3790** | **0.8444** | **0.1464** | **18.9249** | **0.6697** | **0.3299** | 23.3380 | 0.8251 | 0.1921 |

Table 12: Quantitative results on a subset of the Places2 dataset, with varying first mask ratios ranging from 40% to 60%, and a fixed second mask ratio of 20%.

| | | Method | Original-First Inpainting Metrics | | | Original-Second Inpainting Metrics | | | First-Second Inpainting Metrics | | |
|---|---|---|---|---|---|---|---|---|---|---|---|
| | | | PSNR | SSIM | LPIPS | PSNR | SSIM | LPIPS | PSNR | SSIM | LPIPS |
| Second Inpainting Methods | StableDiffusion | DeepFillv2 | 17.7902 | 0.7482 | 0.2735 | 17.1320 | 0.5901 | 0.3919 | **25.5924** | 0.7621 | 0.2026 |
| | | EdgeConnect | 16.6286 | 0.7255 | 0.2859 | 16.1354 | 0.5703 | 0.4030 | 25.2335 | 0.7388 | 0.2258 |
| | | CoModGAN | 16.5925 | 0.7195 | 0.2620 | 16.1611 | 0.5656 | 0.3792 | 24.8675 | 0.7461 | 0.1926 |
| | | StableDiffusion | 15.6794 | 0.6957 | 0.2643 | 15.2555 | 0.5399 | 0.3809 | 25.0807 | **0.7816** | **0.1702** |
| | | LaMa | **18.7100** | **0.7593** | **0.2352** | **17.9365** | **0.6018** | **0.3551** | 25.5705 | 0.7540 | 0.2025 |
| | LaMa | DeepFillv2 | 17.7902 | 0.7482 | 0.2735 | 17.5983 | 0.7148 | 0.3128 | 32.5974 | 0.9665 | 0.0436 |
| | | EdgeConnect | 16.6286 | 0.7255 | 0.2859 | 16.4800 | 0.6920 | 0.3241 | 32.5631 | 0.9663 | 0.0419 |
| | | CoModGAN | 16.5925 | 0.7195 | 0.2620 | 16.4422 | 0.6859 | 0.3005 | 32.4879 | 0.9661 | 0.0417 |
| | | StableDiffusion | 15.6794 | 0.6957 | 0.2643 | 15.5483 | 0.6619 | 0.3021 | 32.4964 | 0.9659 | **0.0412** |
| | | LaMa | **18.7100** | **0.7593** | **0.2352** | **18.4756** | **0.7260** | **0.2740** | 32.6011 | **0.9666** | 0.0419 |
| | DeepFillv2 | DeepFillv2 | 17.7902 | 0.7482 | 0.2735 | 17.5404 | 0.7090 | 0.3203 | **31.3589** | **0.9607** | 0.0525 |
| | | EdgeConnect | 16.6286 | 0.7255 | 0.2859 | 16.4363 | 0.6862 | 0.3314 | 31.3454 | 0.9605 | 0.0503 |
| | | CoModGAN | 16.5925 | 0.7195 | 0.2620 | 16.3993 | 0.6802 | 0.3075 | 31.3348 | 0.9606 | 0.0497 |
| | | StableDiffusion | 15.6794 | 0.6957 | 0.2643 | 15.5095 | 0.6561 | 0.3096 | 31.3306 | 0.9602 | **0.0492** |
| | | LaMa | **18.7100** | **0.7593** | **0.2352** | **18.3982** | **0.7199** | **0.2816** | 31.3252 | 0.9605 | 0.0508 |

Table 13: Quantitative results on a subset of the Places2 dataset, with varying first mask ratios ranging from 40% to 60%, and a fixed second mask ratio of 40%.

| | | Method | Original-First Inpainting Metrics | | | Original-Second Inpainting Metrics | | | First-Second Inpainting Metrics | | |
|---|---|---|---|---|---|---|---|---|---|---|---|
| | | | PSNR | SSIM | LPIPS | PSNR | SSIM | LPIPS | PSNR | SSIM | LPIPS |
| Second Inpainting Methods | StableDiffusion | DeepFillv2 | 17.7902 | 0.7482 | 0.2735 | 16.8276 | 0.5551 | 0.4288 | 23.7716 | 0.7249 | 0.2435 |
| | | EdgeConnect | 16.6286 | 0.7255 | 0.2859 | 15.9004 | 0.5362 | 0.4394 | 23.6027 | 0.7021 | 0.2668 |
| | | CoModGAN | 16.5925 | 0.7195 | 0.2620 | 15.9357 | 0.5316 | 0.4148 | 23.2653 | 0.7080 | 0.2326 |
| | | StableDiffusion | 15.6794 | 0.6957 | 0.2643 | 15.0847 | 0.5067 | 0.4144 | 23.4685 | **0.7431** | **0.2089** |
| | | LaMa | **18.7100** | **0.7593** | **0.2352** | **17.5905** | **0.5677** | **0.3909** | **23.8487** | 0.7174 | 0.2415 |
| | LaMa | DeepFillv2 | 17.7902 | 0.7482 | 0.2735 | 17.3505 | 0.6761 | 0.3523 | 28.6469 | 0.9278 | 0.0867 |
| | | EdgeConnect | 16.6286 | 0.7255 | 0.2859 | 16.2838 | 0.6531 | 0.3626 | 28.6063 | 0.9273 | 0.0837 |
| | | CoModGAN | 16.5925 | 0.7195 | 0.2620 | 16.2436 | 0.6468 | 0.3392 | 28.5275 | 0.9269 | 0.0833 |
| | | StableDiffusion | 15.6794 | 0.6957 | 0.2643 | 15.3792 | 0.6227 | 0.3402 | 28.5544 | 0.9265 | **0.0822** |
| | | LaMa | **18.7100** | **0.7593** | **0.2352** | **18.1764** | **0.6874** | **0.3128** | 28.6547 | **0.9279** | 0.0833 |
| | DeepFillv2 | DeepFillv2 | 17.7902 | 0.7482 | 0.2735 | 17.2145 | 0.6641 | 0.3676 | 27.4044 | **0.9158** | 0.1041 |
| | | EdgeConnect | 16.6286 | 0.7255 | 0.2859 | 16.1834 | 0.6415 | 0.3774 | **27.4083** | 0.9157 | 0.1000 |
| | | CoModGAN | 16.5925 | 0.7195 | 0.2620 | 16.1381 | 0.6345 | 0.3541 | 27.3103 | 0.9149 | 0.0994 |
| | | StableDiffusion | 15.6794 | 0.6957 | 0.2643 | 15.2907 | 0.6112 | 0.3554 | 27.3663 | 0.9150 | **0.0981** |
| | | LaMa | **18.7100** | **0.7593** | **0.2352** | **18.0074** | **0.6752** | **0.3280** | 27.3760 | 0.9158 | 0.1003 |

Table 14: Quantitative results on a subset of the Places2 dataset, with varying first mask ratios ranging from 40% to 60%, and a fixed second mask ratio of 60%.

| | | Method | Original-First Inpainting Metrics | | | Original-Second Inpainting Metrics | | | First-Second Inpainting Metrics | | |
|---|---|---|---|---|---|---|---|---|---|---|---|
| | | | PSNR | SSIM | LPIPS | PSNR | SSIM | LPIPS | PSNR | SSIM | LPIPS |
| Second Inpainting Methods | StableDiffusion | DeepFillv2 | 17.7902 | 0.7482 | 0.2735 | 16.4103 | 0.5225 | 0.4620 | 22.0647 | 0.6914 | 0.2789 |
| | | EdgeConnect | 16.6286 | 0.7255 | 0.2859 | 15.5556 | 0.5034 | 0.4743 | 21.9793 | 0.6680 | 0.3051 |
| | | CoModGAN | 16.5925 | 0.7195 | 0.2620 | 15.5958 | 0.4993 | 0.4475 | 21.7146 | 0.6743 | 0.2678 |
| | | StableDiffusion | 15.6794 | 0.6957 | 0.2643 | 14.8211 | 0.4752 | 0.4450 | 21.9188 | **0.7084** | **0.2429** |
| | | LaMa | **18.7100** | **0.7593** | **0.2352** | **17.1162** | **0.5358** | **0.4234** | **22.1772** | 0.6844 | 0.2759 |
| | LaMa | DeepFillv2 | 17.7902 | 0.7482 | 0.2735 | 16.9981 | 0.6285 | 0.3961 | 25.7239 | 0.8801 | 0.1340 |
| | | EdgeConnect | 16.6286 | 0.7255 | 0.2859 | 16.0006 | 0.6051 | 0.4056 | 25.6769 | 0.8792 | 0.1298 |
| | | CoModGAN | 16.5925 | 0.7195 | 0.2620 | 15.9597 | 0.5988 | 0.3819 | 25.6171 | 0.8787 | 0.1285 |
| | | StableDiffusion | 15.6794 | 0.6957 | 0.2643 | 15.1398 | 0.5746 | 0.3824 | 25.6559 | 0.8781 | **0.1272** |
| | | LaMa | **18.7100** | **0.7593** | **0.2352** | **17.7628** | **0.6401** | **0.3555** | **25.7547** | **0.8805** | 0.1281 |
| | DeepFillv2 | DeepFillv2 | 17.7902 | 0.7482 | 0.2735 | 16.7838 | 0.6118 | 0.4182 | **24.5723** | **0.8633** | 0.1585 |
| | | EdgeConnect | 16.6286 | 0.7255 | 0.2859 | 15.8273 | 0.5878 | 0.4276 | 24.4957 | 0.8618 | 0.1539 |
| | | CoModGAN | 16.5925 | 0.7195 | 0.2620 | 15.7817 | 0.5808 | 0.4042 | 24.4307 | 0.8611 | 0.1524 |
| | | StableDiffusion | 15.6794 | 0.6957 | 0.2643 | 14.9893 | 0.5580 | 0.4047 | 24.5025 | 0.8615 | **0.1505** |
| | | LaMa | **18.7100** | **0.7593** | **0.2352** | **17.4978** | **0.6221** | **0.3785** | 24.5624 | 0.8625 | 0.1535 |

