# OpenReview forum: "A Novel Evaluation Framework for Image Inpainting via Multi-Pass Self-Consistency"
_ICLR.cc/2024/Conference — ICLR 2024 Conference Withdrawn Submission_

### Official Review · Reviewer_R8hK · 2023-10-13

**Soundness:** 1 poor
**Presentation:** 2 fair
**Contribution:** 1 poor
**Rating:** 1
**Confidence:** 4

**Summary:**

This study claims two primary concerns in image inpainting evaluation:

1. The potential bias when depending solely on unmasked images for evaluation.
2. The potential unavailability of unmasked images in practical situations.

The authors introduce a two-stage pipeline, incorporating two distinct mask schemes, to evaluate image inpainting methods.

**Strengths:**

- The proposed method is straightforward and intuitive.
- The evaluation framework aligns more closely with human perception than existing metrics such as MUSIQ and PAR.

**Weaknesses:**

**Regarding Motivation:**

1. The authors argue that the primary flaw in current evaluation methods (MSE, PSNR, SSIM, LPIPS, FID) is their reliance on unmasked images. This assertion lacks rigor. For instance, metrics like MSE and PSNR do not inherently depend on unmasked pixels, especially considering the common post-inpainting operation: $\hat{\mathbf{X}} = \hat{\mathbf{X}} * (1 - \mathbf{M}) + \mathbf{X} \odot \mathbf{M}$.
2. The assertion in Section 1, Para. 2, that unmasked images might not always be accessible in practical scenarios. It seems unlikely to happen under current inpainting settings. Could the authors provide several examples to support this claim?
3. The definition of "potential bias," as mentioned in Sec. 1, Para. 3, remains unclear. Even with the aid of Figure 1, the nature of this bias is not immediately evident.

**Regarding Results:**

1. As per Table 4, the values derived from the proposed metrics are strikingly similar, particularly for smaller mask ratios. This is in contrast to human scores, which exhibit significant variance.

**Questions:**

No.
In conclusion, the motivation behind this study appears nebulous and lacks precision. The proposed methodology does not offer substantial insights. Given these concerns, I am inclined to strongly recommend rejection.

---

### Official Review · Reviewer_ZP4Y · 2023-10-31

**Soundness:** 1 poor
**Presentation:** 2 fair
**Contribution:** 1 poor
**Rating:** 3
**Confidence:** 5

**Summary:**

This submission proposes a self-supervised image inpainting metric for assessing the inpainting performance without the corresponding unmasked images. The contributions lie in the new evaluation framework involving the intrinsic self-consistency of the corrupted image. However, the motivation of "what I cannot create, I do not understand" is unreasonable in this task and the evaluation is one-sided.

**Strengths:**

1) This submission focuses on the problem of assessing the inpainting performance without the corresponding images, which is an interesting and unsolved problem.
2) The intrinsic self-consistency may be a potential way to explore this problem.

**Weaknesses:**

1) The motivation is to achieve a high level of consistency through a profound understanding of the generated content. However,  the design of patch masks is to focus on the restoration of the low-level details, which often rely on the local similarity priors, rather than the high-level understanding. Moreover,  the performance of the second inpainted images depends on the second inpainting network, which highly affects the evaluation.
2) The validation in Table 1 is unreasonable. First, original-first and original-second have the same task, which is to recover the content similar to the original image. Thus, their performance of different methods is consistent, but not the persistence of biases in the metric. Second, for the difference in First-Second, it may be due to the different inpainting mechanisms of CNN-based methods and Diffusion-based methods. It cannot show the superiority of the proposed metric. Third, it is unclear the meanings of 1-to-2, 2-to-0, 2-to-0.
3) Overall, the contributions are limited, and more reasonable motivation and theoretical and experimental validations are needed.

**Questions:**

Please refer to the above weaknesses.

**Details Of Ethics Concerns:**

It needs a user study to assess the alignment of the proposed metric with human perception.

---

### Official Review · Reviewer_rSjt · 2023-10-31

**Soundness:** 2 fair
**Presentation:** 3 good
**Contribution:** 2 fair
**Rating:** 3
**Confidence:** 4

**Summary:**

This paper addresses the evaluation of image inpainting algorithms based on the self-consistency of images. The proposed evaluation framework contains two stages of inpainting, where the first inpainting is to recover the original image under the normal mask and the second inpainting is to recover the first inpainted image under the multiple patch masks, then the perceptual metric (like LPIPS) is implemented on the first inpainted image and the second inpainted images (corresponding to different multiple patch masks) as the evaluation results of the first inpainting method.

**Strengths:**

- The analysis about the current evaluation of image inpainting is useful.
- The idea to provide a comprehensive evaluation of image inpainting is helpful.

**Weaknesses:**

- The analysis about why the proposed framework works for evaluating the image inpainting algorithms is confusing. It seems that the framework tries to use the first inpainting method to recover the masked regions from the unmasked regions of the original image with a normal mask, and then employ another random inpainting method to recover the patch-masked regions from the unmasked regions of the first inpainted image with multiple patch masks that are different from the first normal mask. But why it works for better evaluating the first inpainting method is not clear.
- The evaluation on the proposed benchmark is not convincing. The experiments are insufficient to validate the efficacy of the proposed metric, since not only the dataset but also the method are not comprehensive enough. And also some details are not clear, for example, what is the second inpainting method for these experiments.
- The ablation study is not sufficient. There are many components that may influence the performance but are not analyzed, for example, what about to choose different second inpainting method, what about to choose different sub-metric, what about to choose different dataset, what about to choose different mask type, etc.
- It seems that the Perceptual Metric in Figure 2 should connect the Second Inpainted Images with the Inpainted Image but not the Images Masked by Second Masks.

**Questions:**

The idea of this work is good but the paper still needs to be revised a lot, please check the weaknesses for reference.

---

### Official Review · Reviewer_yYxo · 2023-11-01

**Soundness:** 2 fair
**Presentation:** 2 fair
**Contribution:** 3 good
**Rating:** 5
**Confidence:** 3

**Summary:**

Based on the property of self-consistency in the assumption of most image inpainting pipelines, the authors propose an innovative framework to benchmark image inpainting methods without the need for (1) reference images and (2) additional training. The experiments demonstrate that the proposed framework aligns with human evaluation better than baseline metrics.

**Strengths:**

- 1) The author discuss an interesting topic of evaluation framework on image inpainting methods.

**Weaknesses:**

- 1) The presentation in Sec. 4 does not highlight the contribution of the paper. See Q1 and the Suggestion.
- 2) The robustness of the evaluation framework is not sound. See Q2, and Q3.

**Questions:**

## Questions
- 1) How does Tab. 1 strengthen the claim that **First-Second** is better than the other two choices (**Original-First** and **Original-Second**)? The claim is already justified by the fact that the original/reference images are often not available.
To me, Tab. 1 only demonstrates that **First-Second** has a different score distribution than **Original-First** and **Original-Second**. A bad evaluation metric can also generate a different score distribution. Is there any assumption that StableDiffusion is the best image inpainting among these methods?

- 2) In Fig. 4, the inpainting results of **Noise=0.1** seems to be more natural to me than **Blend**, yet their evaluation in Tab. 3 suggests otherwise, i.e. the scores of **Blend** is lower than **Noise=0.1**. Why is this the case?

- 3) In Tab. 4, **Stable Diffusion** is far superior to other methods in terms of human evaluation, yet the proposed evaluation metric shows only a fraction of difference on **First Mask Ratio=0.0** (27.60 vs 28.xx). Is the proposed evaluation metric useful when comparing between two inpainting sota with competitive or close performance? Is the proposed evaluation metric only useful when there exist one absolutely superior candidate in terms of performance?

- 4) In Limitation, the fact the proposed framework requires more time is briefly discussed. How does it compare to the evaluation time of other baselines like MUSIQ and PAR?

- 5) In Fig. 2, it seems like the LPIPS is computed between **Image Masked by Second Masks** and **Second Inpainted Image** instead of **First Inpainted Image** and **Second Inpainted Image**. However the caption and also Eq. 1 suggest that process *calculate the perceptual metric between the inpainted
images and the corresponding re-inpainted image*. Is this correct?

- 6) How does the evaluation framework perform when normal masks are used instead of patch masks? To prove certainly that using **patch masks** is the better option, I believe this experiment is necessary.

- 7) How does Sec. 4.2 and Tab. 2 show that **Second mask ratio=40%** is the best choice?


## Suggestions
- 1) I suggest the author to move Sec. 4.5 to the 4.2 instead and highlight this subsection since this subsection is the most interesting and important experiment to demonstrate the contribution of this paper: that the proposed framework aligns with human evaluation better than previous metrics.

- 2) I suggest to move the discussions of Fig. 3f to the Sec. 4 as they are a part of ablation studies.